# The Immunological and Epidemiological Effectiveness of Pediatric Single-Dose Vaccination against Hepatitis A 9 to 11 Years after Its Implementation in the Tyva Republic, the Russian Federation

**DOI:** 10.3390/vaccines12080907

**Published:** 2024-08-10

**Authors:** Maria A. Lopatukhina, Karen K. Kyuregyan, Anastasia A. Karlsen, Fedor A. Asadi Mobarkhan, Ilya A. Potemkin, Vera S. Kichatova, Olga V. Isaeva, Lyudmila Yu. Ilchenko, Anna A. Saryglar, Mikhail I. Mikhailov

**Affiliations:** 1Laboratory of Viral Hepatitis, Mechnikov Research Institute of Vaccines and Sera, 105064 Moscow, Russia; marialopatukhina@yandex.ru (M.A.L.); a.carlsen@yandex.ru (A.A.K.); 1amfa@bk.ru (F.A.A.M.); axi0ma@mail.ru (I.A.P.); vera_kichatova@mail.ru (V.S.K.); isaeva.06@mail.ru (O.V.I.); ilchenko-med@yandex.ru (L.Y.I.); michmich2@yandex.ru (M.I.M.); 2Laboratory of Molecular Epidemiology of Viral Hepatitis, Central Research Institute of Epidemiology, 111123 Moscow, Russia; 3Chumakov Federal Scientific Center for Research and Development of Immunobiological Products of Russian Academy of Sciences, 108819 Moscow, Russia; 4Kyzyl Hospital of Infectious Diseases, 667003 Kyzyl, Tyva Republic, Russia; anna_kyzyl@mail.ru

**Keywords:** hepatitis A, hepatitis A vaccine, single-dose vaccination, epidemiology, incidence, public health

## Abstract

Since 2012, universal single-dose HAV vaccination in children aged 3 years and older has been implemented in the Tyva Republic, a region of the Russian Federation. The aim of this prospective non-interventional observational single-center study was to determine the immunological and epidemiological effectiveness of single-dose vaccination against hepatitis A 9 to 11 years after its implementation. The anti-HAV IgG antibodies were determined in two independent cohorts of children who were vaccinated with a single dose of monovalent pediatric inactivated vaccine (HAVRIX^®^ 720 EU) in Tyva in 2012 and recruited 9 years (Year 9 Cohort) and 11 years (Year 11 Cohort) after immunization. The seroprotection rates defined as anti-HAV antibody concentrations ≥10 mIU/mL reached 99.4% (95% CI: 98.2–99.9% [501/504]) in the Year 9 Cohort, but decreased significantly to 75.4% (95% CI: 73.0–77.6% [1006/1335]) in the Year 11 Cohort (*p* < 0.0001). The anti-HAV geometric mean concentrations decreased from 1446.3 mIU/mL (95% CI: 1347.1–1545.4 mIU/mL) in the Year 9 Cohort to 282.6 mIU/mL (95% CI: 203.8–360.8, *p* < 0.0001) in the Year 11 Cohort. The HAV vaccination program resulted in zero rates of hepatitis A incidence in the Tyva Republic since 2016. However, the limited monitoring of HAV RNA in sewage and environmental samples demonstrated the ongoing circulation of both the regional epidemic strain of HAV genotype IA and another genotype IA strain imported recently from other parts of the Russian Federation, probably due to subclinical infections in non-vaccinated children under 3 years of age. Taken together, these data indicate the effectiveness of the single-dose HAV vaccination strategy but suggest the need to expand the vaccination program to include children aged 12 months and older to achieve maximum effectiveness.

## 1. Introduction

Hepatitis A is an acute liver disease caused by the hepatitis A virus (HAV), a member of the *Picornaviridae* family, which is preventable through vaccination. A universal mass vaccination (UMV) strategy in toddlers is shown to be beneficial in regions with HAV endemicity transition from high to intermediate [1]. The standard immunization schedule for the inactivated HAV vaccine consists of two doses given at a six-month interval, and it has proven to be highly immunogenic, providing a protective antibody response expected to last for decades [2]. To decrease the economic burden associated with UMV and to improve the vaccination coverage rates, a universal single-dose HAV vaccination strategy was first implemented in toddlers in Argentina in 2005 [3], followed by Brazil in 2014 [4]. This single-dose schedule has proved to be effective in terms of both a long-lasting immunological response and a decrease in the symptomatic disease burden [5]. However, more data from different regions are needed to understand the persistence of immunity and the effectiveness of single-dose immunization at the population level.

Since 2012, universal single-dose HAV vaccination in children aged 3 years and older (up to 18 years) has been implemented in the Tyva Republic, a region of the Russian Federation that has had HAV incidence levels 10–15-fold higher than the national average [6]. Tyva is located in southern Siberia and borders Mongolia to the south. According to the Human Development Index (HDI), Tyva is the least developed region in Russia [7]. In August 2012, in Tyva, the vaccination campaign began with a monovalent pediatric inactivated vaccine (HAVRIX^®^ 720 EU) being given to children aged 3–8 years. By the end of 2012, 65,097 children received single-dose immunization, resulting in 87.4% coverage among children aged 3–8 years [6]. Since then, single-dose vaccination against hepatitis A has been part of Tyva’s regional immunization schedule for children aged 3 years and older, up to 18 years.

Currently, the UMV strategy is not implemented nationwide in the Russian Federation. With the exception of Tyva and three other regions in which a standard two-dose pediatric HAV immunization has been introduced, HAV vaccination is recommended only for professional risk groups and travelers, as well as for outbreak control. This approach has been shown to have little impact on herd immunity to HAV in Russia [8].

A previous study of the five-year immunological effectiveness of the single-dose HAV immunization in Tyva demonstrated the presence of anti-HAV antibody concentrations of ≥10 mIU/mL in 98.0%, 93.5%, and 91.1% of the children tested one month, one year, and five years after single-dose immunization, respectively [6]. This study also showed a rapid decline in the registered annual incidence rates in children under 18 years of age from 450 to 860 per 100,000 in the pre-vaccination years to 7.5 per 100,000 in this age group and to 3.2 per 100,000 in the total population one year after the start of vaccination, which was further reduced to a zero-incidence level in the region by 2016. These data have confirmed that single-dose vaccination is an effective method of bringing hepatitis A under control in a short period of time in a highly endemic region. However, further monitoring of the incidence rates and antibody levels is needed to determine the sustainability of the observed effect. The existing data on the long-term immunological and epidemiological effectiveness of the hepatitis A pediatric single-dose vaccination strategy have only been generated in Argentina [9] and Brazil [10], and no data are available currently from Asian or European countries.

The single-dose UMV strategy is expected to provide long-lasting protective immunity against HAV that leads to a rapid and sustainable drop in the disease incidence in both the vaccinated and general population and a significant decrease in or even the cessation of HAV circulation, which is usually reflected by a decrease in or even the absence of detectable HAV RNA in wastewater. To evaluate the long-term immunological and epidemiological effectiveness of the single-dose vaccination in the Tyva Republic, three key parameters were assessed in this study: (i) the prevalence of the protective levels of anti-HAV antibodies in the vaccinated population nine and eleven years after the single-dose immunization; (ii) the effect of the vaccination on the reported annual incidence rates of hepatitis A in the study region in both the vaccinated and non-vaccinated populations; (iii) the circulation of HAV in this region after the implementation of universal child single-dose vaccination based on the results of the monitoring of HAV RNA in sewage and open bodies of water.

## 2. Materials and Methods

### 2.1. Study Design

A graphic representation of this prospective non-interventional observational single-center surveillance study is shown in Figure 1. The anti-HAV IgG antibodies were determined at two time points in two independent cohorts of children who were vaccinated with a single dose of monovalent pediatric inactivated vaccine (HAVRIX^®^ 720 EU) in Tyva in 2012 and had available HAV vaccination records. The sample collection and anti-HAV IgG antibody testing were performed in two cohorts, in 2021, i.e., nine years after immunization (Year 9 Cohort), and in 2023, i.e., eleven years after immunization (Year 11 Cohort). The study involved one visit per subject. However, since the study comprised two independent time points (Year 9 and Year 11), the same subject who participated in the Year 9 Cohort could have been enrolled in the Year 11 Cohort. In that case, the subject was enrolled as a new subject and was not associated with their participation in the Year 9 survey. Blood samples were collected at the Kyzyl Hospital of Infectious Diseases, and then sera were shipped to the Mechnikov Research Institute for Vaccines and Sera for anti-HAV antibody testing. To supplement the data on the immunogenicity of single-dose HAV vaccination with epidemiology data, the annual hepatitis A incidence rates reported in the Tyva Republic in 2013–2023 were analyzed in comparison to the pre-vaccination period (2001–2012). During the study period, 2021–2023, samples of sewage, sources of drinking water (springs and wells), and open bodies of water were collected for HAV RNA monitoring. Sampling was performed twice per year, in June and October.

### 2.2. Study Cohorts and Blood Sampling

The population sample size was calculated using the effect size calculation [11] with a significance level of 5% and a power of 99%. The minimal sample size was calculated to be 470 for both cohorts, rounded up to 500. In 2021, 504 children (aged 11–18 years, median age 14 years) who were vaccinated with one dose of HAV vaccine were recruited for the study. This group of participants is referred to as the Year 9 Cohort. The female/male ratio in this cohort was 1:0.8, and the urban/rural population ratio was 1:1.2. In 2023, an independent cohort of 500 children vaccinated with one dose of the HAV vaccine in 2012 was recruited. However, due to the high demand from the parents of vaccinated children for voluntary sampling and anti-HAV antibody testing, an additional 835 children were recruited and tested, comprising a total sample size of 1335 participants in the Year 11 Cohort, aged 13–18 years with a median age of 15 years, a 1:1.1 female/male ratio, and a 1:1.4 urban/rural population ratio. An additional sensitivity analysis [11] was performed to assess the impact of the increasing sample size on statistical power. The power was >99% when the sample size was increased to 1335 participants.

The study was conducted in accordance with the principles expressed in the World Medical Association Declaration of Helsinki regarding ethical medical research involving human subjects. The signed informed consent of each participant’s parent was obtained before recruitment. Additionally, the signed informed consents of the subjects of age 15 to 18 years were obtained in addition to the signed informed consent from their parents.

The vaccination status of the participants was retrieved from the medical records (individual vaccination cards, Form No.63) stored at children’s polyclinics. In compliance with the study protocol, all the enrolled participants met the following inclusion criteria:•Subjects whose parents complied with the requirements of the protocol.•Written informed consent obtained from the subject and/or subject’s parents.•Children who received one dose of the HAV monovalent pediatric inactivated vaccine (HAVRIX^®^ 720 EU) with available HAV vaccination records.•Not more than 6 months of deviation from the study’s time point: the time between the date of vaccination and the date of blood sampling was between 8.5 and 9.5 years in the Year 9 Cohort and between 10.5 and 11.5 years in the Year 11 Cohort.

In compliance with the study protocol, the following were excluded from enrollment: children in care, children who received two doses of the vaccine or received hepatitis A vaccines other than Havrix, and children with a known history of hepatitis A infection before vaccination.

Serum samples with a volume of ca. 3 mL were obtained from each enrolled participant and shipped using a cold chain to the core lab in Moscow, the Mechnikov Research Institute for Vaccines and Sera. All the sera were coded and aliquoted, and the 0.5 mL aliquots were stored at −70 °C until testing. Following the anti-HAV IgG antibody testing, the children with antibody concentrations below 20 mIU/mL were offered a booster dose of vaccine. No additional measurement of antibodies was performed for these children.

### 2.3. Anti-HAV Testing

Total anti-HAV antibodies were tested in the sera of the vaccinated children using two commercially available quantitative immunoassays: Elecsys^®^ Anti-HAV (Roche, Mannheim, Germany) on a cobas e 411 analyzer and an ELISA Vectohep A-IgG kit (Vector-Best, Novosibirsk, Russia). Both the assays have similar performance characteristics and provide results expressed in International Units per liter (IU/L). The testing was performed according to the instructions provided by the manufacturers of the respective kits. Seropositivity was defined as antibody levels of ≥20 mIU/mL. The alternative or surrogate seroprotection cutoff level was defined as an anti-HAV antibody concentration of ≥10 mIU/mL, as this concentration was previously reported as minimally protective in humans [12]. All the samples with anti-HAV antibody concentrations above the upper limit of the quantification range of the test were diluted and repeatedly tested. The final concentrations were obtained by multiplying the result by the dilution factor.

### 2.4. Incidence Analysis

Data on the reported annual hepatitis A incidence rates were retrieved from the database of the Russian Federal Service for Surveillance on Consumer Rights Protection and Human Wellbeing (Rospotrebnadzor). The hepatitis A incidence rates in Tyva from 2001 to 2023 were analyzed in comparison to the national average rates for the following population groups: (i) children aged 0–14 years; (ii) children aged 0–17 years; (iii) total population. Additionally, the hepatitis A incidence rates from 2013 to 2023 in the regions neighboring Tyva (the Republic of Buryatia, the Republic of Khakassia, the Republic of Altai, Irkutsk Region, and Krasnoyarsk Territory) were analyzed. To assess the possible changes in the incidence of other enteric infections during the HAV vaccination period (2013–2023), the incidence rates of enterovirus infections and shigellosis in the total population of Tyva were analyzed.

### 2.5. HAV RNA Testing in Environmental Samples

In total, 337 samples of sewage and water from different sources were collected, including 120 sewage samples from Kyzyl city, the capital of Tyva; 82 samples from sources of drinking water, such as springs and wells; and 135 samples from open basins near settlements, including lakes and rivers at places popular for swimming and sunbathing. The samples were collected at six time points, twice a year, in July and in October, 2021–2023. The sampling was performed at the same locations at each time point.

The volume of each sample was 2 L. The concentration of the water samples was carried out immediately after the sampling using the commercially available kit “Virosorb-M” (Bioservice, Russia) according to the manufacturer’s protocol. The method of processing is based on the concentration of negatively charged viral particles on magnetic particles coated with polymer silicon dioxide modified with amino groups. The volume of the resulting concentrated sample was 2 mL. The concentrated samples were shipped using a cold chain to the core lab in Moscow for the subsequent nucleic acid extraction and HAV RNA testing. The isolation of total nucleic acids was performed from the 1 mL concentrated samples using the MagNA Pure Compact Nucleic Acid Isolation Kit I—Large Volume (Roche Applied Science, Mannheim, Germany). HAV RNA was detected using a polymerase chain reaction combined with reverse transcription (RT-PCR) with primers for the VP1/2A region of the virus genome using the previously described protocol [13]. All the HAV-positive samples were sequenced to obtain the genetic information on strains circulating in Tyva. For this purpose, the amplified fragments of the HAV genome were sequenced in a 3130 Genetic Analyzer (ABI, Foster City, CA, USA) automatic sequencer using the BigDye Terminator v3.1 Cycle Sequencing Kit according to the manufacturer’s protocol. The HAV sequences were subjected to phylogenetic analysis using the maximum likelihood (ML) method in the IQ-TREE software (v.2.3.6), together with the HAV sequences obtained in different areas of Russia, including Tyva, in different years.

### 2.6. Statistical Analysis

Data analysis was performed using graphpad.com. The statistical analysis included the calculation of the geometric mean concentration (GMC) for anti-HAV antibody concentrations, the calculation of a 95% confidence interval (95% CI), and the assessment of the significance of the differences in the mean values between the study cohorts using Fisher’s exact test (for categorical data) and an unpaired t-test (for continuous data) with the significance threshold set to *p* < 0.05.

## 3. Results

### 3.1. Persistence of Anti-HAV Antibodies after Single-Dose Vaccination

The results of the anti-HAV antibody testing in the Year 9 and Year 11 Cohorts are summarized in Table 1.

In the Year 9 Cohort, protective anti-HAV antibody concentrations (≥20 mIU/mL) were detected in 99.4% (95% CI: 98.2–99.9% [501/504]) of the children tested. Among the 501 seropositive samples, 440 samples contained anti-HAV antibodies with concentrations in the range of 20 to 6000 mIU/mL (minimum—30 mIU/mL, maximum—1632 mIU/mL).

In the Year 11 Cohort, the proportions of the serum samples with anti-HAV antibody concentrations ≥20 mIU/mL were significantly lower compared to the Year 9 Cohort, at 71.8% (95% CI: 69.3–74.1% [958/1335]) (*p* <0.0001). The proportions of the samples with low anti-HAV concentrations, between 10 and 19 mIU/mL, were 0% and 3.6% in the Year 9 and Year 11 Cohorts, respectively. Given that the possible protective threshold level could be as low as 10 mIU/mL, we calculated the proportion of samples with anti-HAV antibody concentrations of ≥10 mIU/mL in each study cohort. The proportion of such samples in the Year 11 Cohort was significantly lower than that in the Year 9 Cohort (Table 2).

In the Year 9 Cohort, 61 out of the 504 samples (12.1%) contained anti-HAV antibodies with a concentration above 6000 mIU/mL, indicating a possible boosted antibody response (Table 1). In the Year 11 Cohort, the proportion of the samples with anti-HAV antibody concentrations above 6000 mIU/mL was significantly lower (8.9%, *p* = 0.0432).

The proportion of the samples with anti-HAV concentrations below 20 mIU/mL in the Year 9 Cohort was 0.6% (95% CI: 0.1–1.8%, [3/504]). In fact, all these samples contained anti-HAV in concentrations below 10 mIU/mL (Table 1). The proportion of the samples with anti-HAV concentrations below 20 mIU/mL was significantly higher in the Year 11 Cohort, 28.2% (95% CI: 25.9–30.7% [377/1335], *p* < 0.0001). If we consider an anti-HAV concentration of ≥10 mIU/mL as a protective threshold level, the proportion of children without protective antibody levels was also significantly higher in the Year 11 Cohort compared to the Year 9 Cohort (24.6% vs. 0.6%, *p* < 0.0001) (Table 1).

The samples with anti-HAV concentrations of <10 mIU/mL and >6000 mIU/mL were excluded from the calculation of the GMC. The anti-HAV antibody GMC values for the study cohorts are shown in Table 3. The anti-HAV antibody GMC value eleven years after the single-dose vaccination was significantly lower compared to that observed in the Year 9 Cohort (*p* < 0.0001, unpaired *t*-test).

### 3.2. Hepatitis A Incidence Analysis

The registered incidence of any infectious diseases in Russia is reported by Rospotrebnadzor for three categories of people: the total population, children aged 0–14 years, and children aged 0–17 years. The hepatitis A annual incidence rates in Tyva are shown in Figure 2 for these three categories in comparison to the respective national averages.

The hepatitis A incidence rates in Tyva in the pre-vaccination period (2001–2012) were the highest in the country, with the majority of the cases registered in children and adolescents under 18 years. In 2020–2023, during the study period, no cases of hepatitis A were registered in Tyva; the same was observed for 2016–2020. Interestingly, the incidence dropped after the start of the vaccination campaign not only in children aged 0–14 years (Figure 2B), i.e., the age group that includes vaccinated children, but also in older children (Figure 2C) and in the total population (Figure 2A).

Next, we analyzed the hepatitis A incidence rates during the vaccination period (2013–2023) in the regions neighboring Tyva to confirm that the drop in incidence in this particular territory did not result from a decrease in HAV circulation in the whole region. Almost every year, with the exception of the period of the COVID-19 pandemic, in all the regions neighboring Tyva, there were registered incidences of hepatitis A, with pronounced increases in some years (Table 4). Thus, against the background of the zero or almost-zero incidence of hepatitis A in Tyva during the vaccination period, substantial incidence rates were registered continuously in the neighboring regions.

To assess whether the observed drop in hepatitis A incidence in Tyva could be possibly associated with the general improvement in sanitation and the consequent decrease in the burden of enteric infections, we additionally analyzed the incidence rates of enterovirus infections and shigellosis in 2013–2023 in Tyva (Table 5). The high annual rates of these enteric infections indicate the continued poor sanitation in the region, which is associated with the sustained risk for the transmission of communicable diseases. Moreover, these data clearly indicate that the hepatitis A incidence drop observed in 2013–2023 was not associated with improved sanitation.

### 3.3. HAV RNA Monitoring in Sewage and Environment Samples

The data on HAV RNA detection in the sewage and water samples are summarized in Table 6. HAV RNA was detected in 7 out of the 337 samples tested, including 2 out of the 120 (1.7%) sewage samples, 2 out of the 82 (2.4%) sources of drinking water, and 3 out of the 135 (2.2%) samples from open bodies of water.

One HAV-containing sample was collected in June 2021 in the lake near the beach that is popular among local residents. Another positive sample was obtained in October 2022 from sewage collected in Kyzyl city. Five HAV RNA-positive samples were identified in October 2023, including one from sewage collected in Kyzyl city, two samples from springs in rural areas, and two samples from the river downstream from where treated sewage from Kyzyl city is discharged.

All the HAV-positive samples were sequenced. The results of the phylogenetic analysis are shown in Figure 3.

The HAV isolates detected in 2021–2022 belong to the genotype IA and are grouped together with the sequences obtained in Tyva in 2008–2010, before the start of vaccination, i.e., they refer to a local epidemic strain. Meanwhile, the HAV sequences isolated from sewage and other environmental samples in 2023 belong to another cluster of genotype IA sequences that were isolated in 2019–2023 in different parts of the Russian Federation, including the neighboring Irkutsk Region. Thus, the HAV sequences isolated in Tyva in 2023 are indicative of the recent importation of infection to this region from the other parts of the Russian Federation.

## 4. Discussion

The primary objective of this study was to assess the immunological effectiveness of single-dose HAV vaccination, i.e., the long-term persistence of seroprotection among children who received one dose of the inactivated HAV vaccine in Tyva. Although anti-HAV IgG antibody concentrations of ≥20 mIU/mL are widely used as the seroprotection cutoff in vaccine licensing and clinical studies, an antibody level of ≥10 mIU/mL is considered to be seroprotective [5]. According to WHO’s position on anti-HAV humoral immunity, the absolute lower limit of protective antibody level has not been determined, and usually, the range of 10–20 mIU/mL is considered as protective depending on the immunoassay used for detection [12]. The recent papers on HAV single-dose vaccination use a cut-off antibody level of ≥10 mIU/mL [9,10]. Thus, to have comparable data, we assessed the level of seroprotection using both the cutoff values in two pediatric cohorts, nine and eleven years after the single-dose immunization. In the Year 9 Cohort, the estimated effectiveness of one HAV vaccine dose was 99.4% regardless of the seroprotection cutoff applied. These data are consistent with the 97.4% seroprotection level reported in the vaccinated children up to 9 years of age following single-dose vaccination at the age of one year in Argentina, where the same cutoff of ≥10 mIU/mL was applied [14]. The level of seroprotection in the Year 9 Cohort was similar to the 91% seroprotection rate observed earlier in Tyva in children five years after single-dose vaccination [6]. However, we observed a significant reduction in the seroprotection rates in the Year 11 Cohort to 71.8% when the ≥20 mIU/mL cutoff was applied and to 75.4% with a cutoff of ≥10 mIU/mL. Correspondingly, the GMCs in the Year 11 Cohort were significantly lower compared to the Year 9 Cohort, indicating a decrease in humoral immunity over time.

The level of HAV seroprotection observed in our study 11 years after the single-dose immunization was substantially lower than the 93% rate observed in a cohort of 27 children followed 12 years after single-dose immunization with the inactivated HAV vaccine in Argentina [9]. However, de Brito and coauthors reported detectable anti-HAV IgG antibodies in 64% of children after 6 to 7 years of single-dose vaccination with the inactivated HAV vaccine [10]. Taken together, these data suggest that the humoral immunity to HAV following a single-dose vaccination might decrease faster compared to the standard two-dose schedule, which provides seroprotection in >90% of vaccinated children for up to 15 years [1,15]. This observation is confirmed by the data from the study that directly compared the duration of humoral immunity to HAV 10 years after single-dose and two-dose vaccination and demonstrated significant differences both in the seroprotection rates (71.9% versus 96.3%) and the GMCs (26.0 mIU/mL versus 82.1 mIU/mL) [16]. Thus, those who received single-dose immunization with inactivated HAV vaccine in childhood might need booster vaccination in adulthood, considering the lack of real-world or mathematical modeling data on the decades-lasting duration of humoral immunity following this vaccination schedule. Moreover, although standard two-dose vaccination is believed to provide decades-long or even life-long immunity to HAV, rare cases of infection in adult individuals at risk who were vaccinated in childhood are described, with evidence of immune escape-driven virus evolution [17]. These cases, albeit rare, suggest that the combination of vanishing vaccine-induced immunity and a high virus dose might result in breakthrough HAV infection. Nevertheless, the waning humoral immunity might not necessarily indicate the lack of protection against HAV. A recent study on HAV-specific T-cell response in children up to 12 years following single-dose vaccination demonstrated the presence of memory CD4+ and CD8+ T-cell responses in 53.8% and 26.9% of seronegative children, respectively [9]. Likewise, the production of interferon gamma in peripheral blood mononuclear cells (PBMCs) stimulated with the HAV VP1 antigen was demonstrated in 32.4% of the seronegative children 6 to 7 years after single-dose vaccination with the inactivated vaccine, indicating the cell-mediated immune memory [10]. Nevertheless, all the children who had anti-HAV IgG antibodies in our study below 20 mIU/mL were recommended a booster dose of the vaccine, considering that CD4+ and CD8+ memory cells may prevent disease but may not prevent infection, allowing the virus to replicate in the presence of a low concentration of antibodies. Moreover, the high proportion of individuals with low or very low anti-HAV antibody concentrations in the population may hypothetically prompt the selection of immune-escaping virus variants. Taken together, these data suggest the need to ensure a significant proportion of individuals with protective antibody levels to limit the virus evolution, even though the emergence of new serotypes is not a characteristic of HAV [18]. In this regard, the pediatric single-dose HAV vaccination strategy might need to be supplemented with a booster dose in adulthood, at least in groups at risk.

Interestingly, significant proportions of children, 8.9% to 12.1%, depending on the cohort, had anti-HAV concentrations above 6000 IU/L. There are two possible explanations for such a large proportion of samples with abnormally high anti-HAV concentrations in the vaccinated children. First, and the most likely, is the vaccination of children previously exposed to HAV, as a screening for antibodies to HAV was not performed prior to vaccination. The history of hepatitis A prior to vaccination was an exclusion criterion in our study, but some participants could have had a subclinical infection. In this case, high levels of anti-HAV antibodies could be associated with post-infection immune response. The second explanation is the exposure to HAV after the vaccination. However, the latter is highly debatable, as no proof of possible transient subclinical HAV infection in vaccinated individuals has been reported so far, and no evidence of the “natural boosting” of HAV immunity has been documented [5]. Nonetheless, the rate reports on HAV breakthrough infections in vaccinated individuals accompanied by virus variant selection [17,19] do not permit to fully exclude such possibility. Regardless of the cause, such samples with abnormally high anti-HAV concentrations were excluded from the GMC calculation, as they were highly likely not related to single-dose immunization.

In addition to the immunological effectiveness, we assessed the epidemiological effectiveness of the single-dose HAV vaccination program in Tyva. The implementation resulted in a sharp decline in hepatitis A incidence rates in both vaccinated children and non-vaccinated adolescents and adults. From 2016 to the present, no single hepatitis A case has been reported in Tyva, making this particular region the first territory in the Russian Federation to be free of symptomatic hepatitis A. The absence of registered hepatitis A cases is apparently not related to underreporting since there is mandatory reporting of hepatitis A in Russia by both laboratories and clinicians. The public health definition of confirmed hepatitis A includes any confirmed positive anti-HAV IgM antibody or positive HAV RNA. Moreover, the zero incidence rates in Tyva are not the result of a global decline in HAV circulation in Russia, since hepatitis A cases were registered regularly during the past decade in the regions neighboring Tyva. Furthermore, the disappearance of symptomatic HAV infections in Tyva was not associated with a general sanitary improvement, since the annual incidence rates of other enteric infections, both viral and bacterial, have remained high in the region.

The possible explanation for the reduction in hepatitis A incidence in the total population may be the fact that the highest incidence rates in the pre-vaccination period in the region were observed in children under 14 years, who obviously served as a main source of infection. Thus, vaccine-induced HAV immunity in children together with the high rates of infection-induced immunity among adults reported in this region [8] provide a sufficient level of herd immunity and significantly reduce the opportunity for HAV circulation. Likewise, a significant and rapid decline in the incidence rates in all the age groups followed the implementation of a single-dose UMV strategy in Argentina and Brazil [20,21,22].

It is expected that a significant reduction in symptomatic HAV infections would be accompanied by a decrease in the HAV detection rates in sewage. However, the three-year monitoring of HAV RNA in the sewage and various open bodies of water in Tyva confirmed the persistence of virus shedding. Moreover, the phylogenetic analysis confirmed that the HAV sequences isolated from the water samples in Tyva in 2021 and 2022 belonged to the same group of strains that were detected in the HAV patients in this region in 2008, in the pre-vaccination period, suggesting the stable circulation of this particular variant of the virus. However, in 2023, another HAV strain was identified in the sewage and environmental samples that belonged to the group of strains identified in 2019–2023 in different regions of Russia, including the Irkutsk Region that borders Tyva. This finding confirms the importation of a new strain into Tyva. It is impossible to conclude whether the UMV strategy has resulted in a decrease in the number of sewage samples positive for HAV RNA since no such monitoring was performed in Tyva before our study. However, in 2002–2004, the HAV antigen testing of sewage samples was performed in Tyva, resulting in 12.5%-63.3% positivity rates depending on the study year [23]. Considering the average proportion of HAV RNA-positive samples in the current study was below 2.5%, one might assume that vaccination resulted in a significant decrease in virus shedding in the region. Interestingly, the environmental monitoring of HAV RNA in Argentina performed following the introduction of the single-dose HAV UMV strategy in 2005 demonstrated the continuous detection of HAV RNA in sewage in the studies conducted in 2009–2010 and in 2017–2022 [24,25]. However, unlike in Tyva, the detection of HAV RNA in the wastewater samples in Argentina correlated with the cases of acute hepatitis A [25]. A possible explanation for the continued shedding of the virus in Tyva and its detection in environmental samples and wastewater despite the absence of registered cases of acute hepatitis A may be associated with the age at which the vaccine is given to children. Obviously, the isolation of the virus in the absence of reported disease cases most probably indicates the presence of asymptomatic infection. Most often, HAV infection is asymptomatic in children under 5 years of age [26]. Typically, HAV vaccination is recommended for children aged 12 months or older [27]. However, in Tyva, single-dose vaccination has been introduced for children aged 3 years and older, primarily in order to not exclude the only domestically produced HAV vaccine licensed for children of that age. Thus, children under 3 years of age are not immunized and remain susceptible to HAV. Therefore, the most likely hypothesis explaining the persistence of HAV circulation in the absence of symptomatic cases is the virus transmission among children under 3 years of age. Our data on continuing HAV circulation despite a decade of the UMV program in Tyva strongly suggest that immunization should be started at the age of 12 months to diminish the vaccination gaps.

This study has two limitations. First, this is a single-center study. However, participants from different districts of Tyva were recruited for this study, both from urban and rural settings, providing an adequate representation of the child population of the study region. Second, the HAV environmental surveillance was not comprehensive in terms of both the sampling design and HAV RNA detection methodology. Indeed, monthly sampling with larger volumes of surface and drinking water samples would provide more accurate data on the presence of HAV in the environment and the picture of the circulating strains. The study utilized an RT-PCR assay targeting the VP1/2A region that is widely used for HAV genotyping but is less sensitive than the standard diagnostic kits targeting the 5′-UTR region. However, as HAV surveillance was not a primary goal of the study, we opted for this less sensitive detection method to obtain only the environmental samples that would yield HAV sequences. Nevertheless, even the less sensitive analysis demonstrated the presence of HAV RNA in the environmental samples of different types in Tyva, and made it possible to identify the change in circulating virus strains.

## 5. Conclusions

Our data demonstrated the immunological and epidemiological effectiveness of pediatric single-dose HAV vaccination in Tyva. The seroprotection rate was as high as 99.4% nine years after the single-dose vaccination in childhood but significantly decreased in the cohort tested eleven years after immunization. The drop in HAV seroprotection rates observed between nine and eleven years after the single-dose vaccination was accompanied by a significant decrease in the anti-HAV geometric mean values. These data indicate that serum anti-HAV antibodies induced by a single-dose immunization vanish over time, with significantly lower levels at eleven years after vaccination than at nine years. However, 75.4% of the children still had protective levels of anti-HAV antibodies eleven years after the single-dose immunization. The UMV strategy based on a single dose of inactivated HAV vaccine resulted in zero rates of hepatitis A incidence in the Tyva Republic. However, the monitoring of HAV RNA in the sewage and environmental samples demonstrated the ongoing circulation of both the regional epidemic strain of the virus and the strain imported recently from other parts of the Russian Federation, probably due to subclinical infections in non-vaccinated children under 3 years of age. Taken together, these data indicate the effectiveness of the single-dose HAV vaccination strategy but suggest the need to expand the vaccination program to include children aged 12 months and older to achieve maximum effectiveness.

## Figures and Tables

**Figure 1 vaccines-12-00907-f001:**
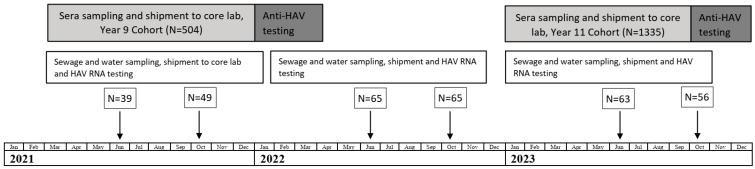
Study design and timeline. N indicates the number of samples.

**Figure 2 vaccines-12-00907-f002:**
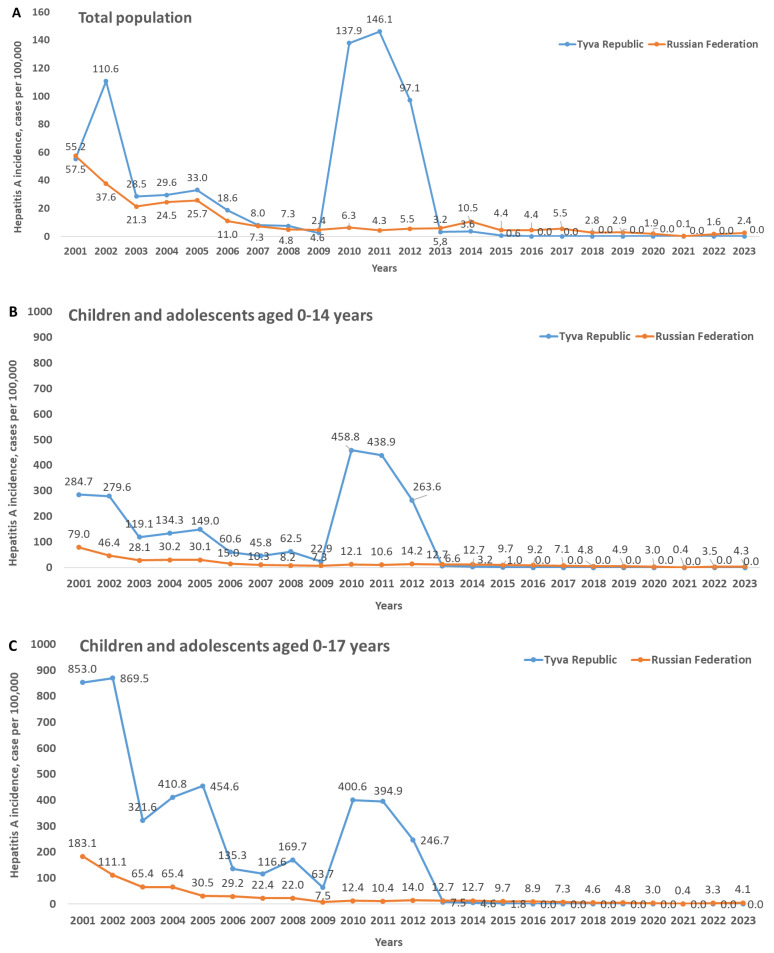
Hepatitis A annual incidence rates from 2001 to 2023 in Tyva and in the Russian Federation on average among the total population (**A**), children aged 0–14 years (**B**), and children aged 0–17 years (**C**).

**Figure 3 vaccines-12-00907-f003:**
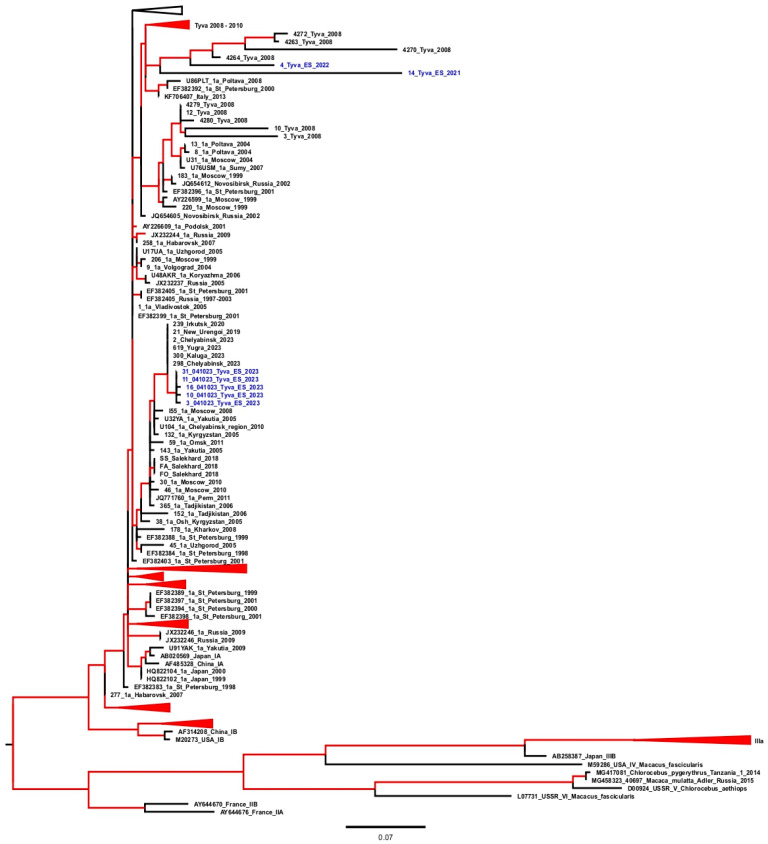
Maximum likelihood phylogenetic tree for the HAV VP1/2A sequences. For each sequence, the number in the GenBank database, the country, the city (in the case of Russian sequences), and the year of isolation are indicated. Sequences from the environmental samples collected in the Tyva Republic are shown in blue with the indicated year of isolation. The tree branches shown in red have a posterior probability of >90%.

**Table 1 vaccines-12-00907-t001:** Distribution of the serum anti-HAV antibody concentrations in the samples collected 9 and 11 years after the single-dose vaccination.

Anti-HAV Concentrations	Year 9 Cohort, n = 504		Year 11 Cohort, n = 1335	
	Number of Samples	Detection Rate, % (95% CI)	Number of Samples	Detection Rate, % (95% CI)
0–9 mIU/mL	3	3/504, 0.6% (0.1–1.8%)	329	329/1335, 24.6% (22.4–27.0%)
*p* *	<0.0001			
10–19 mIU/mL	0	0/504, 0%	48	48/1335, 3.6% (2.7–4.7%)
*p* *	<0.0001			
20–6000 mIU/mL	440	440/504, 87.3% (84.1–89.9%)	839	839/1335, 62.9% (60.2–65.4%)
*p* *	<0.0001			
>6000 mIU/mL	61	61/504, 12.1% (9.5–15.3%)	119	119/1335, 8.9% (7.5–10.6%)
*p* *	0.0432			

* comparing data from the Year 11 Cohort with data from the Year 9 Cohort (Fisher’s exact test).

**Table 2 vaccines-12-00907-t002:** Proportion of the samples with anti-HAV antibody concentrations of ≥10 mIU/mL collected 9 and 11 years after the single-dose vaccination.

Anti-HAV Concentrations	Year 9 Cohort, n = 504		Year 11 Cohort, n = 1335	
	Number of Samples	Detection Rate, % (95% CI)	Number of Samples	Detection Rate, % (95% CI)
≥10 mIU/mL	501	99.4% (98.2–99.9%)	1006	75.4% (73.0–77.6%)
*p* *	<0.0001			

* comparing data from the Year 11 Cohort with data from the Year 9 Cohort (Fisher’s exact test).

**Table 3 vaccines-12-00907-t003:** Anti-HAV antibody geometric mean concentrations in samples collected 9 and 11 years after single-dose vaccination.

Cohort	Anti-HAV GMC, mIU/mL (95% CI)	*p* *
Year 9 Cohort	1446.3 (1347.1–1545.4)	<0.0001
Year 11 Cohort	282.6 (203.8–360.8)	

* comparing data from the Year 11 Cohort with data from the Year 9 Cohort (unpaired *t*-test).

**Table 4 vaccines-12-00907-t004:** Hepatitis A incidence rates in the total population of Tyva and in neighboring regions in 2013–2023.

Year	Hepatitis A Incidence, Cases per 100,000					
	Tyva Republic	Republic of Buryatia	Republic of Khakassia	Republic of Altai	Irkutsk Region	Krasnoyarsk Territory
2013	3.20	5.35	4.13	3.37	5.61	11.29
2014	3.55	6.48	34.55	2.39	9.37	34.93
2015	0.60	1.44	23.99	2.37	9.83	13.86
2016	0.00	2.25	6.54	0.00	5.46	7.91
2017	0.00	2.96	2.80	1.87	8.91	5.17
2018	0.00	1.02	3.91	1.85	4.94	2.44
2019	0.00	1.63	1.68	32.50	5.46	2.78
2020	0.00	0.1	1.31	5.47	3.93	2.02
2021	0.00	0.00	0.19	0.00	0.04	0.00
2022	0.00	0.71	0.75	0.00	1.73	1.89
2023	0.00	0.82	0.38	0.00	1.57	1.93

**Table 5 vaccines-12-00907-t005:** Enterovirus infections and shigellosis incidence rates in the total population of Tyva, 2013–2023.

Year	Incidence, Cases per 100,000	
	Enterovirus Infections	Shigellosis
2013	0.65	197.6
2014	10.00	207.48
2015	5.14	157.82
2016	93.36	119.26
2017	22.24	101.36
2018	19.55	70.64
2019	174.56	49.83
2020	28.23	10.13
2021	1.23	0.00
2022	77.83	9.96
2023	113.73	7.42

**Table 6 vaccines-12-00907-t006:** Results of HAV RNA monitoring in the sewage and water samples.

Sampling Time Point	Sewage Samples		Sources of Drinking Water (Springs and Wells)		Open Bodies of Water	
	Number of Tested Samples	Number of Positive Samples (%)	Number of Tested Samples	Number of Positive Samples (%)	Number of Tested Samples	Number of Positive Samples (%)
June, 2021	16	0 (0%)	9	0 (0%)	14	1 (7.1%)
October, 2021	16	0 (0%)	13	0 (0%)	20	0 (0%)
June, 2022	24	0 (0%)	15	0 (0%)	26	0 (0%)
October, 2022	24	1 (4.2%)	15	0 (0%)	26	0 (0%)
June, 2023	24	0 (0%)	15	0 (0%)	24	0 (0%)
October, 2023	16	1 (6.3%)	15	2 (13.3%)	25	2 (8.0%)
Total	120	2 (1.7%)	82	2 (2.4%)	135	3 (2.2%)

## Data Availability

The data presented in this study are available in this article.

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
