# Peer review of "The Immunological and Epidemiological Effectiveness of Pediatric Single-Dose Vaccination against Hepatitis A 9 to 11 Years after Its Implementation in the Tyva Republic, the Russian Federation"

_vaccines, 2024, doi:10.3390/vaccines12080907_

Round 1

Reviewer 1 Report

Comments and Suggestions for Authors

This is a nice ms with valuable information for decision-making public health Authorities. The work is well executed and presented. However, there are some points that should be discussed to achieve a broader interpretation of the results. Additionally, some minor points may also be improved.

Critical points

1The most critical methodological part is the environmental surveillance:

2 1) Sampling design is not correct. At least a monthly sample would be expected to have a real scenario of what circulates among the population.

3 2) Two liters of sewage is correct, but for surface and drinking water is insufficient. The rule of thumb is a volume as large as possible with a minimum of 20L. Since this cannot be corrected at this point, it should be stated in the discussion section this important limitation.

4 3) In the discussion it is mandatory to emphasize the limitation imposed using the VP1-2A target for HAV detection. The VP1-2A region is the target for HAV genotyping but it has a very low sensitivity compared with methods based on the 5’NCR, which indeed are much more conserved among different HAV genotypes and thus show a broader detection. The best way to proceed is: HAV detection using the 5’NCR and using the positive samples for typing using the VP1-2A region as described in reference 23 of the ms.

TThe most important epidemiological issues to be included in the discussion are:

51) The data presented supports that serum anti-HAV antibodies induced with a single-dose immunization vanishes over time, with significantly lower levels at 11 years after vaccination than at 9 years after vaccination.

6 2) The main question thus is what would happen at 25 years after vaccination when vaccinated children will be 28 yo. For instance, some hepatitis A cases have been described in MSM patients previously vaccinated with two doses during their childhood (doi: 10.1016/j.ebiom.2018.11.023.). CD4+ and CD8+ memory cells may prevent disease but may not prevent infection, allowing the virus to replicate in the presence of low concentration of antibodies.

7 3) A low concentration of antibodies if patients get infected may prompt the selection of escaping viruses, and this is something at least to be mentioned in the discussion in a pros and cons paragraph.  

 iMinor points:

 1) Lines 28-31 should be re-written in view of the comments on the environmental surveillance.

9 2) Lines 52 and 61. Children 3 years and older, how older?

1 3) Figure 2. Panels B and C would be better interpreted if using the same scale in the incidence axis.

1 4) Lines 367-369. This is indeed very debatable and should be taken into consideration the previous comments.

Author Response

We thank the reviewer for their meaningful comments.

Comment 1

The most critical methodological part is the environmental surveillance:

1) Sampling design is not correct. At least a monthly sample would be expected to have a real scenario of what circulates among the population.

2) Two liters of sewage is correct, but for surface and drinking water is insufficient. The rule of thumb is a volume as large as possible with a minimum of 20L. Since this cannot be corrected at this point, it should be stated in the discussion section this important limitation.

3) In the discussion it is mandatory to emphasize the limitation imposed using the VP1-2A target for HAV detection. The VP1-2A region is the target for HAV genotyping but it has a very low sensitivity compared with methods based on the 5’NCR, which indeed are much more conserved among different HAV genotypes and thus show a broader detection. The best way to proceed is: HAV detection using the 5’NCR and using the positive samples for typing using the VP1-2A region as described in reference 23 of the ms.

Response

We agree that the HAV environmental surveillance was not comprehensive in terms of both sampling design and HAV RNA detection methodology. Indeed, monthly sampling with larger volumes of surface and drinking water samples would provide more accurate data on presence of HAV in environment and the picture of circulating strains. The study utilized RT-PCR assay targeting VP1/2A region that is widely used for HAV genotyping but is less sensitive than standard diagnostic kits targeting 5’-UTR region. However, as the HAV surveillance was not a primary goal of the study, we opted this less sensitive detection method to obtain only environmental samples that would yield HAV sequences. Unfortunately, we are unable to test environmental samples using PCR targeting 5’-UTR, as these samples are not available any longer.  Nevertheless, even less sensitive analysis employed in our study demonstrated the presence of HAV RNA in environmental samples of different types in Tyva, and made it possible to identify the change in virus circulating strains. We added this point as a limitation of the study in Discussion (lines 463-476 in revised manuscript).

Comment 2

The most important epidemiological issues to be included in the discussion are:

The data presented supports that serum anti-HAV antibodies induced with a single-dose immunization vanishes over time, with significantly lower levels at 11 years after vaccination than at 9 years after vaccination.

Response

We stressed this important point in Conclusion (lines 483-485 in revised manuscript).

Comment 3

The main question thus is what would happen at 25 years after vaccination when vaccinated children will be 28 yo. For instance, some hepatitis A cases have been described in MSM patients previously vaccinated with two doses during their childhood (doi: 10.1016/j.ebiom.2018.11.023.). CD4+ and CD8+ memory cells may prevent disease but may not prevent infection, allowing the virus to replicate in the presence of low concentration of antibodies.

Response

We added these considerations to Discussion (lines 359-368 in revised manuscript).

Comment 4

A low concentration of antibodies if patients get infected may prompt the selection of escaping viruses, and this is something at least to be mentioned in the discussion in a pros and cons paragraph. 

Response

We added these point to Discussion (lines 377-386 in revised manuscript).

Comment 5

Lines 28-31 should be re-written in view of the comments on the environmental surveillance.

Response

We would like to keep this sentence in Abstract as it is, as it does not contradict the data from our study, even if the environmental surveillance was not comprehensive. To reflect the limitation of the HAV monitoring, we described it as a “limited” (line 28 in revised manuscript).

Comment 6

Lines 52 and 61. Children 3 years and older, how older?

Response

Up to 18 years. We added this information (lines 53 and 62 in revised manuscript).

Comment 7

Figure 2. Panels B and C would be better interpreted if using the same scale in the incidence axis.

Response

We modified Y-axis scale for panels B and C, now it is the same, up to 1000.

Comment 8

Lines 367-369. This is indeed very debatable and should be taken into consideration the previous comments.

Response

We added considerations regarding the HAV breakthrough infections in vaccinated individuals accompanied by virus variant selection to this part (lines 398-400 in revised manuscript).

Reviewer 2 Report

Comments and Suggestions for Authors

Congratulations on the great working and writing. There are no repairs for me to make to the current manuscript. 

Author Response

Comment

Congratulations on the great working and writing. There are no repairs for me to make to the current manuscript.

Response

We are very grateful to the reviewer for the positive comments on the manuscript.

Reviewer 3 Report

Comments and Suggestions for Authors

Lopatukhina et al. present a study interesting and important in the field of public health. It investigates antibody titers in children vaccinated against hepatitis A, measured 9 and 11 years after inoculation. The antibody titers were associated with the effectiveness of the vaccination, which involved a single dose. The article is well-founded, well-written, and clear in its methodology and results. It presents a thorough discussion and acknowledges some limitations in establishing hypotheses related to the decrease in hepatitis A cases post-vaccination, along with other significant aspects.

I only have a couple of comments:

According to the cited source (ref. 12), "the absolute lower limit of protective antibody level has not been determined," and usually the range of 10-20 mIU/ml is considered, "depending on the immunoassay used for detection." Some important considerations are mentioned at the beginning of section 4.1 (ref. 12). While using the cut-off value of 10 mIU/ml is not incorrect, I recommend clarifying this data in the text.

In Table 1, I recommend adding the number of data used for the calculations. In Table 6, I suggest adding the totals for each column.

Author Response

We thank the reviewer for thorough analysis of the manuscript and positive comments.

Comment 1

According to the cited source (ref. 12), "the absolute lower limit of protective antibody level has not been determined," and usually the range of 10-20 mIU/ml is considered, "depending on the immunoassay used for detection." Some important considerations are mentioned at the beginning of section 4.1 (ref. 12). While using the cut-off value of 10 mIU/ml is not incorrect, I recommend clarifying this data in the text.

Response

We added this clarification to Discussion (lines 331-335 in revised manuscript).

Comment 2

In Table 1, I recommend adding the number of data used for the calculations. In Table 6, I suggest adding the totals for each column.

Response

Done.

Reviewer 4 Report

Comments and Suggestions for Authors

This manuscript analyzed antibody responses in 9-year and 11-year-olds after universal single-dose HAV vaccination in the Tyva Republic, a region of the Russian Federation. Immunological responses to the hepatitis A vaccine are analyzed to show the effectiveness of the single-dose hepatitis A vaccine in the study subjects. The manuscript is well written; the findings of the study are clear and easy to understand.

Review comments

In the study design section of Materials and Methods, information regarding the name of the facility where the sample collection was performed is missing.

In line 125, the 11-year cohorts need to have information on the ages of the study subjects.

The limitations of the study need to be described in the Discussion. Immunological responses to the hepatitis A vaccine were analyzed in a single center. 

Author Response

We are very grateful to the reviewer for the comments on the manuscript.

Comment 1

In the study design section of Materials and Methods, information regarding the name of the facility where the sample collection was performed is missing.

Response

We added the name of the facility where the sample collection was performed to Materials and Methods section (lines 108-110 in revised manuscript).

Comment 2

In line 125, the 11-year cohorts need to have information on the ages of the study subjects.

Response

We added data on the ages of the study subjects in 11-year cohort (line 129 in revised manuscript).

Comment 3

The limitations of the study need to be described in the Discussion. Immunological responses to the hepatitis A vaccine were analyzed in a single center.

Response

We added this point to the limitations of the study in Discussion, which include single-center study and not comprehensive environmental surveillance (lines 463-476 in revised manuscript).